# Use of extracorporeal membrane oxygenation for eCPR in the emergency room in patients with refractory out-of-hospital cardiac arrest

**L. Christian Napp**[1]**, Carolina Sanchez Martinez**[1]**, Muharrem Akin**[1]**, Vera Garcheva**[1]**, Christian Kühn**[2]**, Johann Bauersachs**[1]**, Andreas Schäfer**[1]*

**1** Klinik für Kardiologie und Angiologie, Medizinische Hochschule Hannover, Hannover, Germany, **2** Klinik für Herz-, Thorax-, Transplantations- und Gefäßchirurgie, Medizinischen Hochschule Hannover, Hannover, Germany

\* Schaefer.andreas@mh-hannover.de

**Data Availability Statement:** There are legal restrictions on sharing the data publicly, as it contains potentially identifying patient information

## Abstract

### Background

Out-of-hospital cardiac arrest (OHCA) refractory to conventional high-quality cardiopulmonary resuscitation (CPR) may be rescued by extracorporeal CPR (eCPR) using veno-arterial extracorporeal membrane oxygenation (V-A ECMO). Even when trying to identify eCPR candidates based on criteria assumed to be associated with a favourable neurological outcome, reported survival rates are frequently below 10%.

### Methods

All patients undergoing implantation of V-A ECMO for eCPR between January 2018 and December 2019 (N = 40) were analysed (age 53±13 years; 75% male). Patients with refractory OHCA and potentially favourable circumstances (initial shockable rhythm, witnessed arrest, bystander CPR, absence of limiting comorbidities, age <75 years) were transported under mechanical chest compression. Candidates for eCPR should have a pH ≥6.9, arterial lactate ≤15 mmol/L and time-to-ECMO should be ≤60 minutes.

### Results

Overall 30-day survival was 12.5%, with 3 of 5 survivors having a favourable neurological outcome (cerebral performance category (CPC) 1 or 2), representing 7.5% of the total eCPR population. No patient selected for eCPR met all pre-defined criteria (median of unfavourable criteria: 3). Importantly, time-to-ECMO most often (39/40) exceeded 60 minutes (mean 102 ±32 min.), and lactate was >15mmol/L in 30 out of 40 patients. Moreover, 22 out of 40 patients had a non-shockable rhythm on the first ECG.

and specific dates. However, other researchers who meet the criteria for access to confidential data may submit requests to the Medizinische Hochschule Hannover Ethics Committee (Ethikkommission@mh-hannover.de).

**Funding:** The study was partly supported by the Clinical Research Group (KFO) 311 of the Deutsche Forschungsgemeinschaft to JB. There was no additional external or internal funding received for this study.

**Competing interests:** AS has received lecture and proctoring honoraria from Abiomed and ZOLL, LCN has received lecture honoraria from Maquet, Abbott, Zoll and Orion as well as lecture, consulting and proctoring honoraria from Abiomed. All other authors declare no conflict of interest. This does not alter our adherence to PLOS ONE policies on sharing data and materials.

**Abbreviations:** CPC, cerebral performance category; (c)CPR, (conventional) cardiopulmonary resuscitation; eCPR, extracorporeal cardiopulmonary resuscitation; ECMO, extracorporeal membrane oxygenation; HACORE, HAnnover Cooling REgistry; OHCA, out-of-hospital cardiac arrest; ROSC, return of spontaneous circulation.

## Conclusions

Despite our intention to select patients with potentially advantageous circumstances to achieve acceptable eCPR outcomes, the imminent deadly consequence of withholding eCPR obviously prompted individual physicians to perform the procedure also in presumably more unfavourable settings, resulting in similar mortality rates of eCPR as reported before.

## Introduction

In patients with out-of-hospital cardiac arrest (OHCA), two major targets have to be addressed: providing high-quality cardiopulmonary resuscitation (CPR) to achieve return of spontaneous circulation (ROSC) as quickly as possible, and re-establishing optimal oxygen supply and end-organ perfusion to prevent brain damage and multi-organ failure [1, 2]. Recommendations in resuscitation care primarily focus on conventional CPR (cCPR) techniques and the management after ROSC. However, there are numerous, often younger, patients with refractory OHCA, i.e. who do not achieve ROSC after 10–15 minutes of cCPR. Therefore, a more invasive and aggressive treatment regimen has been proposed to prevent unsuccessful cCPR resulting in imminent death. This approach consists of implantation of veno-arterial extracorporeal membrane oxygenation (V-A ECMO) during cCPR to rapidly provide circulatory support and sufficient blood oxygenation at the same time, termed extracorporeal CPR (eCPR) [3–8].

Initial reports suggested that implementation of eCPR in standard pathways for OHCA treatment can improve survival [9]. However, more recent and larger analyses revealed that overall survival was not higher than with cCPR, even if implementing eCPR in a metropolitan emergency response system in Paris over a seven years period [10]. While 4% of OHCA patients received eCPR, outcome was not improved by this strategy and survival remained lower than 10%, comparable to patients treated with cCPR. The authors suggested identifying eCPR candidates by an initially shockable rhythm and transient ROSC during cCPR. Encouraged by the authors' initial report three years before [9], we introduced an eCPR algorithm for patients with refractory OHCA in our region. Selection criteria were adapted from the German consensus statement on eCPR [11], and were applied with the intention to select qualified candidates for eCPR to improve outcomes. Here, we report on our experience with this strategy, the feasibility of selection, and efficacy of this approach over a two-year period.

## Methods

The HAnnover Cooling REgistry (HACORE) is a prospective observational registry approved by the ethics committee at Hannover Medical School (#3567–2017) in accordance with the Declaration of Helsinki. The ethics committee approved the analysis as reported in the present manuscript. Written informed consent was obtained from legal guardians during the unconscious period and re-consented by survivors after gaining consciousness. HACORE includes anonymized data from all OHCA patients treated at our cardiac arrest centre with a standardized protocol including therapeutic hypothermia and mechanical circulatory support if required [1, 12–15]. In the present analysis, all OHCA patients undergoing implantation of V-A ECMO for eCPR between January 2018 and December 2019 were analysed. During that period, N = 40 patients were available for analysis.

Positive selection criteria to initiate eCPR were refractory arrest, an initial shockable rhythm, witnessed arrest, rapid commencement of bystander CPR, absence of limiting comorbidities, and biological age <75 years. For candidate patients we recommended to transport them under mechanical chest compression using the LUCAS device (Stryker Medical, Portage MI, USA). The emergency medical service had been informed and trained on identifying those criteria. Upon hospital admission, we suggested that candidates for eCPR should ideally have a minimum pH ≥6.9, an arterial lactate ≤15 mmol/L and a time-to-ECMO ≤60 minutes. However, due to the lack of robust prospective data and the imminent deadly consequence of excluding patients based on a single parameter, those variables were considered ideal but not essential.

Decision for or against eCPR was made by interdisciplinary agreement between cardiologists and cardiac surgeons, who convened at patient arrival in the emergency department together with anaesthetists and the emergency department staff. V-A ECMO was implanted by cardiac surgeons in the emergency department by percutaneous femoral access, using 15 to 17 F arterial and 22 to 24 F venous cannulas. Antegrade perfusion sheaths were implanted in the cathlab under fluoroscopic guidance.

Patients were treated according to local standards of the cardiac arrest centre including multiple measures of post-resuscitation care as described earlier [1]. In brief, patients underwent routine computed tomography, coronary angiography including intervention where indicated, optional left ventricular unloading with Impella pumps, neuroprotective hypothermia for at least 24 hours, invasive hemodynamic monitoring, and continuous neuromonitoring. To perform therapeutic hypothermia on intensive care unit (ICU), an intravascular cooling catheter (Coolguard Quattro®, ZOLL Medical, San Jose, CA, USA) was placed in the right femoral vein if hypothermia could not be maintained by ECMO. An active cooling device is generally chosen to select and maintain a constant target temperature of 32°C for 24–48 hours followed by controlled rewarming (0.25°C per hour) and normothermia for another 72 hours.

Patients were followed up for the period of their hospital stay and data were extracted from the electronic hospital patient data management system. Discharge letters from rehabilitation facilities were collected when patients had been transferred to such an institution. Neurological outcome was assessed using cerebral performance category (CPC).

Baseline characteristics are presented as frequencies (n) and percentages (%) for categorical variables, means ± standard deviation (SD) for normally distributed continuous variables, or median and interquartile ranges (IQR) for non-normally distributed continuous variables. Normally distributed variables were compared by Student´s t-test and Mann-Whitney test for nonparametric data, respectively. All group comparisons of continuous measures were performed using Wilcoxon's test, whereas chi-square or Fisher's exact test were used to assess categorical data. Cumulative mortality was estimated by Kaplan-Meier method with statistical significance examined by the log-rank test.

Statistical analyses were performed using SPSS Statistics 24 (IBM SPSS Statistics 24). A two-sided p-value of <0.05 was considered statistically significant.

## Results

Between January 2018 and December 2019 N = 40 patients (age 53±13 years; 75% male) underwent V-A ECMO implantation for eCPR (Table 1). The relatively high rate of males might be related to the fact that acute myocardial infarction (AMI) was the most common cause of cardiac arrest and in AMI populations, a ratio of three out of four patients being male is commonly observed.

**Table 1. Patient characteristics.**

|  | All n = 40 | Survivors n = 5 | Non-survivors n = 35 | P-value |
|---|---|---|---|---|
| Baseline characteristics N (%) |  |  |  |  |
| Age (years) | 52.6 ±12.9 | 43.4 ±15.0 | 53.9 ±12.4 | 0.0913 |
| Age ≥75 years | 2 (5%) | 0 (0%) | 2 (6%) | 0.5947 |
| Age ≥65 years | 6 (15%) | 0 (0%) | 6 (17%) | 0.3278 |
| Age ≥55 years | 20 (50%) | 1 (20%) | 19 (54%) | 0.1594 |
| Sex (males) | 30 (75%) | 3 (60%) | 27 (77%) | 0.4206 |
| Primary rhythm |  |  |  | 0.4838 |
| Ventricular fibrillation / tachycardia | 18 (45%) | 3 (60%) | 15 (43%) |  |
| Asystole | 6 (15%) | 2 (40%) | 4 (11%) |  |
| Pulseless electric activity | 16 (40%) | 0 (0%) | 16 (46%) |  |
| Suspected pulmonary embolism | 6 (15%) | 1 (20%) | 5 (14%) | 0.7457 |
| Arrest setting |  |  |  |  |
| Witnessed arrest | 37 (93%) | 4 (80%) | 33 (94%) | 0.2681 |
| Bystander CPR | 33 (83%) | 5 (100%) | 28 (80%) | 0.2827 |
| Professional CPR prior to EMS | 11 (28%) | 2 (40%) | 9 (26%) | 0.5158 |
| Use of mechanical compression device | 23 (58%) | 3 (60%) | 20 (57%) | 0.9068 |
| Time to ECMO (min) | 102 ±32 | 106 ±44 | 101 ±31 | 0.7474 |
| Time to ECMO ≥60 min | 39 (98%) | 5 (100%) | 34 (97%) | 0.7107 |
| Time to ECMO ≥90 min | 28 (70%) | 2 (40%) | 26 (74%) | 0.1237 |
| Time on ECMO (hrs) | 56 ±90 | 145 ±130 | 43 ±78 | **0.0165** |
| Admission laboratory |  |  |  |  |
| pH | 6.94 ±0.14 | 6.87 ±0.04 | 6.95 ±0.15 | 0.2348 |
| pH ≤6,9 | 23 (58%) | 4 (80%) | 19 (54%) | 0.2885 |
| Lactate [mmol/L] | 14.5±1.5 | 13.9±2.5 | 14.5±1.4 | 0.3904 |
| Lactate ≥15 mmol/L | 32 (80%) | 4 (80%) | 28 (80%) | >0.9999 |
| Δ Lactate after 4 hrs [mmol/L] | 3.3 ±3.4 | 3.8 ±1.4 | 3.2 ±3.6 | 0.7180 |
| HCO$_3$ [mmol/L] | 21.2 ±8.5 | 21.5 ±11.1 | 21.2 ±8.3 | 0.9494 |
| blood glucose [mg/dl] | 407 ±127 | 373 ±122 | 412 ±128 | 0.5278 |
| PaO$_2$ [mmHg] | 236 ±188 | 210 ±191 | 240 ±190 | 0.7419 |
| eGFR [ml/min] | 78 ±25 | 99 ±18 | 74 ±25 | **0.0468** |
| Unfavourable criteria present | 3 [IQR 2;4] | 3 [IQR 2;4] | 3 [IQR 2;4] | 0.9112 |

CPR–cardiopulmonary resuscitation; ECMO–extracorporeal membrane oxygenation; eGFR–estimated glomerular filtration rate; EMS–emergency medical service

During that period, overall 30-day survival on eCPR was 12.5%, with 3 out of 5 survivors having a favourable neurological outcome with a cerebral performance category (CPC) of 1 or 2, representing only 7.5% of the total eCPR population. While there was a numerical trend in favour of younger age, primary shockable rhythm and presence of bystander CPR, the trend for time-to-ECMO and admission pH were in the opposite direction; however, the low number of survivors hampered the statistical validity of our analysis. The only significant difference between survivors and non-survivors was detected for baseline renal function (eGFR 99±18 vs. 74±25 ml/min, P = 0.0468), while there was a trend for younger age in survivors (43±15 vs. 54 ±12 years, P = 0.0913).

Although we had consented selection criteria for eCPR initiation as outlined above, we realized that adherence to those was poor. Despite our intention to select patients with potentially advantageous circumstances to achieve acceptable eCPR outcomes, the imminent deadly consequence of withholding eCPR obviously prompted individual physicians to perform the

procedure also in more unfavourable settings. Over the two years period, no patient selected for eCPR met all initially suggested criteria (median of unfavourable criteria: 3 both for survivors and non-survivors). Importantly, time-to-ECMO most often (39/40) exceeded 60 minutes (mean 102±32 min.), and lactate was >15mmol/L in 30/40 patients. Also, 22/40 patients had a non-shockable rhythm at first ECG (Table 2). After retrospectively modifying our criteria to age <75 years, witnessed arrest, bystander CPR, and initial shockable rhythm, still only 13/40 patients fulfilled all criteria. Of the individual criteria, in general, the age limit of 75 years, witnessed arrest and bystander CPR were reasonably well complied with. The aim to implement V-A ECMO within 60 minutes after arrest seemed to be too challenging given the current regional infrastructure. Admission pH and lactate were neither prognostically helpful nor obeyed.

Of 40 patients, 5 survived to hospital discharge (12.5%). Of the five survivors, two fulfilled the modified criteria (survival rate for modified criteria 15.4%), one with good neurological outcome (CPC 1), and one with disadvantageous neurological outcome (CPC 4). On the contrary, one of the patients with good neurological outcome (CPC 2) was a drowning victim with initial asystole following 45 minutes under water and time-to-ECMO of 240 minutes. Only one of the survivors had an initial lactate <15 mmol/L (Table 3).

Of note, no patient with age >55 years, eGFR ≤75 ml/min, or CPR without bystander CPR survived on eCPR with good neurological outcome. Although the presence of a non-shockable rhythm did not preclude survival, the 30 day survival rate was very low (6.8%, Fig 1).

## Discussion

The present study in a cohort of OHCA patients without ROSC found that outcome of eCPR was poor despite prior definition of selection criteria for V-A ECMO use. The reasons may be low adherence to criteria, bad predictive value of the proposed criteria and a rather long median time-to-ECMO.

Survival in OHCA patients remains low, although numerous steps in the chain of rescue have been optimized over time. In recent years, many tertiary centres have started to use eCPR with intention to rescue OHCA patients without ROSC, whose outcome would otherwise have been fatal. Indeed, initial reports of single cases and small series suggested that eCPR may be able to rescue a substantial proportion of patients [16, 17]. However, eCPR is obviously a very resource-intensive strategy and cannot be applied to all OHCA patients, at least not with current devices and in current settings. Therefore, many centres apply predefined criteria for use of eCPR in order to allocate V-A ECMO to patients with a reasonable chance of survival [11, 18]. Indeed, application of predefined criteria for eCPR was associated with increased survival in an urban tertiary centre initiative [9]. However, although criteria such as ventricular fibrillation or bystander CPR are associated with a higher chance of survival, it is challenging to predict survival in a single individual based on those criteria. The surviving patients in our cohort fulfilled only few of the "positive" selection criteria for eCPR. This may in part be explained by difficulties in assessing criteria: asystole may have been fine ventricular fibrillation on a preclinical monitor, bystander CPR may have been of varying efficacy, and now-flow and low-flow times may have been miscalculated. Nevertheless, while there are no single factors predicting good neurological outcome after restoring systemic circulation and gas exchange, we could identify certain clinical conditions that are strongly associated with poor outcome: age >55 years, failure to initiate basic life support immediately after arrest, and impaired renal function (eGFR <75 ml/min) on admission. One of the historically most important predictors of eCPR success is the time from arrest to commencement of ECMO flow (time-to-ECMO), i.e. the sum of no-flow and low-flow time [19]. In our cohort, the predefined criteria may have

**Table 2. Non-favourable conditions.**

| Condition | survivors | | | | | non-survivors | | | | | | | | | | | | | | | | | | | | | | | | | | | | | | | | | | |
|---|---|---|---|---|---|---|---|---|---|---|---|---|---|---|---|---|---|---|---|---|---|---|---|---|---|---|---|---|---|---|---|---|---|---|---|---|---|---|---|---|
| Age >75 years | 0 | 0 | 0 | 0 | 0 | 0 | 0 | 0 | 0 | 0 | 0 | 0 | 0 | 0 | 0 | 0 | 0 | 0 | 0 | 0 | 0 | 0 | 0 | 1 | 0 | 0 | 0 | 0 | 0 | 0 | 0 | 0 | 0 | 0 | 0 | 0 | 0 | 1 | 0 | 0 |
| Age >65 years | 0 | 0 | 0 | 0 | 0 | 0 | 0 | 0 | 0 | 0 | 0 | 0 | 0 | 0 | 0 | 0 | 0 | 1 | 0 | 0 | 0 | 0 | 0 | 1 | 0 | 0 | 0 | 1 | 0 | 0 | 0 | 0 | 1 | 0 | 0 | 0 | 1 | 1 | 0 | 0 |
| Age >55 years | 0 | 0 | 0 | 1 | 0 | 1 | 0 | 0 | 0 | 0 | 1 | 1 | 0 | 1 | 0 | 1 | 0 | 1 | 1 | 0 | 1 | 0 | 1 | 1 | 0 | 1 | 0 | 1 | 0 | 0 | 0 | 1 | 1 | 1 | 1 | 0 | 1 | 1 | 1 | 0 |
| tECMO >60' | 1 | 1 | 1 | 1 | 1 | 1 | 1 | 1 | 1 | 1 | 1 | 1 | 1 | 1 | 1 | 1 | 1 | 1 | 1 | 1 | 1 | 1 | 1 | 1 | 1 | 1 | 1 | 1 | 1 | 1 | 1 | 1 | 1 | 1 | 1 | 1 | 0 | 1 | 1 | |
| tECMO >90' | 1 | 0 | 0 | 0 | 1 | 1 | 1 | 1 | 1 | 1 | 1 | 1 | 1 | 1 | 1 | 1 | 1 | 1 | 0 | 1 | 0 | 0 | 0 | 1 | 0 | 1 | 1 | 0 | 0 | 1 | 1 | 1 | 1 | 1 | 1 | 1 | 0 | 0 | 0 | 1 | 1 |
| unwitnessed | 1 | 0 | 0 | 0 | 0 | 0 | 0 | 0 | 0 | 0 | 0 | 0 | 0 | 0 | 0 | 0 | 0 | 0 | 0 | 0 | 0 | 0 | 0 | 0 | 1 | 0 | 0 | 0 | 0 | 0 | 0 | 0 | 0 | 0 | 0 | 0 | 0 | 1 | 0 | 0 |
| no bystander | 0 | 0 | 0 | 0 | 1 | 0 | 0 | 0 | 0 | 0 | 0 | 0 | 0 | 1 | 1 | 0 | 0 | 0 | 0 | 0 | 0 | 0 | 0 | 1 | 0 | 0 | 0 | 0 | 0 | 0 | 0 | 0 | 1 | 0 | 0 | 1 | 1 | 0 | | |
| non shockable | 0 | 0 | 0 | 1 | 1 | 1 | 1 | 1 | 1 | 1 | 1 | 1 | 1 | 0 | 0 | 0 | 1 | 0 | 0 | 1 | 0 | 0 | 0 | 0 | 1 | 1 | 1 | 0 | 0 | 1 | 1 | 0 | 1 | 1 | 0 | 1 | 0 | 1 | 1 | 0 |
| pH <6.9 | 1 | 1 | 1 | 1 | 0 | 0 | 0 | 0 | 1 | 0 | 0 | 1 | 1 | 0 | 1 | 1 | 1 | 0 | 1 | 1 | 1 | 0 | 0 | 1 | 1 | 0 | 1 | 0 | 1 | 0 | 1 | 1 | 1 | 1 | <6.9 | 0 | 0 | 1 | 0 | 1 | 1 | 1 | 0 |
| Lactate >15 mmol/L | 1 | 1 | 1 | 1 | 0 | 0 | 1 | 1 | 1 | 1 | 0 | 1 | 1 | 1 | 0 | 1 | 1 | 0 | 0 | 1 | 1 | 1 | 0 | 0 | 1 | 1 | 1 | 1 | 1 | 1 | 1 | 1 | 0 | 1 | 1 | 1 | 1 | 1 | 0 | 1 | 1 |
| Σ original factors | 4 | 3 | 3 | 4 | 2 | 3 | 3 | 3 | 4 | 3 | 2 | 4 | 4 | 3 | 3 | 3 | 4 | 1 | 2 | 4 | 3 | 2 | 1 | 3 | 6 | 3 | 4 | 2 | 3 | 4 | 4 | 2 | 3 | 3 | 3 | 4 | 2 | 5 | 5 | 2 |
| modified checklist | 0 | 1 | 1 | 0 | 0 | 0 | 0 | 0 | 0 | 0 | 0 | 0 | 0 | 0 | 0 | 0 | 1 | 0 | 1 | 1 | 0 | 1 | 1 | 1 | 0 | 0 | 0 | 0 | 1 | 1 | 0 | 1 | 0 | 0 | 0 | 0 | 1 | 0 | 0 | 1 |

tECMO–time to initiation of extracorporeal membrane oxygenation

not been as predictive as they might have lost predictive value due to a rather long time-to-ECMO, which could be a surrogate for prolonged low-flow times. This again emphasises the critical role of improving preclinical management of eCPR candidates, and future eCPR programs need to shorten the low-flow-time to a minimum.

**Table 3. Individual characteristics of survivors.**

| | Age, sex | Arrest situation | Initial rhythm | CPR | Lactate, mmol/L | pH | tECMO, minutes | Unfavourable factors | BMIkg/m$^2$ | NSE d2, µg/l | s100b d2, µg/l | admis. Temp. | CPC final |
|---|---|---|---|---|---|---|---|---|---|---|---|---|---|
| #1 | 45, ♂ | Unwitnessed collapse on train, CPR by physician incl. ventilation by laryngeal mask. AMI due to LAD occlusion, PCI 3 stents | VT | BLS by physician | 15 | 6.85 | 113 | 4 | 27.8 | 37 | 0.783 | 33.1 | 2 |
| #2 | 48, ♂ | Witnessed arrest with immediate bystander CPR, History of angina before collapse, in ER switch from manual CPR to LUCAS, pulseless tachycardia with anterior AMI due to LAD occlusion, PCI 1 stent | VF | BLS by lay person | 15 | 6.85 | 82 | 4 | 25.7 | 37 | 0.335 | 33.2 | 1 |
| #3 | 48, ♂ | Witnessed car accident w/o trauma, immediate bystander CPR. AMI due to RCA occlusion, PCI 2 stents | VF | BLS by lay person | 15 | 6.85 | 75 | 3 | 32.1 | 208 | N/A | 33.6 | 4 |
| #4 | 58, ♀ | Witnessed collapse at work following dyspnea, immediate bystander CPR, transport on LUCAS, CT confirmed bilateral central pulmonary embolism | Asystole | BLS by lay person | 15 | 6.85 | 80 | 4 | 23.9 | 648 | 0.57 | 34.1 | 4 |
| #5 | 18, ♀ | Witnessed drowning (lake), 45 minutes under water, transport to local hospital, spoke-to-hub transfer on LUCAS | Asystole | ACLS after rescue | 9.5 | 6.94 | 240 | 2 | 23.4 | 47 | 0.277 | 30.9 | 2 |

ACLS–advanced cardiac life support; AMI–acute myocardial infarction; BLS–basic life support; BMI–body-mass indexCPC–cerebral performance category; CPR–cardiopulmonary resuscitation; CT–computed tomography; ER–emergency room; LAD–left anterior descending coronary artery; NSE–neuron-specific enolase (neuromarker), PCI–percutaneous coronary intervention; RCA–right coronary artery; s100b –S-100b protein (neuromarker); tECMO–time to extracorporeal membrane oxygenation; VF–ventricular fibrillation; VT–ventricular tachycardia

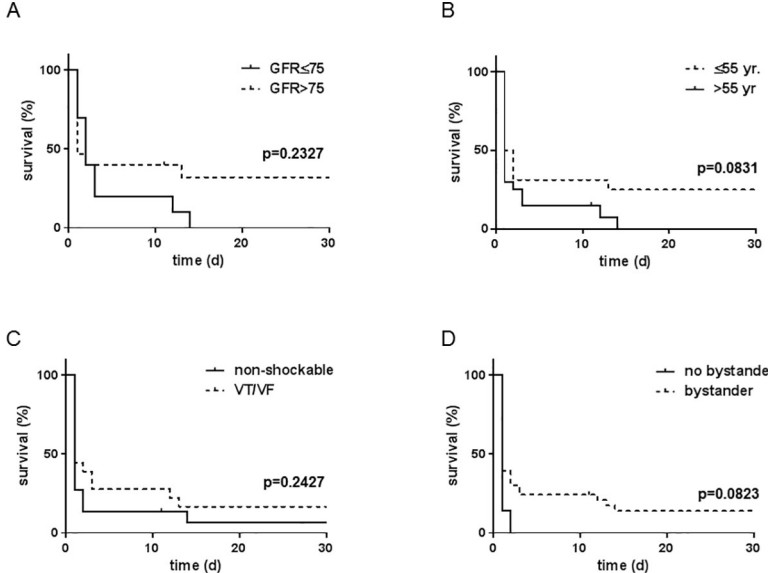

**Fig 1. Survival following extracorporeal cardiopulmonary resuscitation (eCPR).** 30-day survival on eCPR depending on admission renal function (A), age (B), presence of shockable rhythm (C) or bystander CPR (D).

In order to improve the success rate of eCPR, we used reliable information at the time of hospital admission to adjust our criteria retrospectively. While eGFR might be statistically relevant, it is not available at the time of clinical decision making. Lactate, while rapidly available, is not sensitive enough. Once we sequentially excluded all patients with age >60 years (n = 12 non-survivors), absence of bystander CPR (n = 4 younger than 60 years), and PEA (n = 1 remaining non-survivor) we had 23 remaining eCPR patients with 5 survivors (survival rate 22%). A success rate of 20–25% might make the intervention more acceptable in the professional community than almost certain fatality if applied without restrictions. In different healthcare systems, restrictions will furthermore be modified or more stringent depending on the reimbursement of V-A ECMO. It has to be discussed that in our system we are not restricted in everyday clinical life regarding the use of extracorporeal devices.

Our registry has several limitations. While the observational design precludes any causal conclusions, the unselected recruitment in one central metropolitan cardiac arrest centre provides a real-life picture of eCPR for refractory OHCA. In contrast to other regional settings, ours does not include a preclinical eCPR service. Due to the limited sample size, statistically significant differences are only hypothesis-generating.

## Conclusion

Positive outcomes from eCPR are still limited, and only a few patients finally survive until discharge. Since some survivors had rather disadvantageous factors, it is very difficult to predict outcomes in individual patients, which complicates decision making in eCPR. Accordingly, adherence to predefined selection criteria was low. We propose that future work should be focused on optimising preclinical management, as prolonged hypooxygenation might be particularly associated with fatal outcome, irrespective of all in-hospital efforts. Nevertheless, time-to-ECMO alone did not appear as good surrogate for extent of hypooxygenation. In general, the predictive value of all predefined criteria was rather low, so final decision making still relies on individual judgement.

## Acknowledgments

We thank the nursing staff of the catheterization laboratory and cardiology intensive care unit for their continuous support and care in treating OHCA patients.

## Author Contributions

**Conceptualization:** L. Christian Napp, Christian Kühn, Johann Bauersachs, Andreas Schäfer.

**Data curation:** Carolina Sanchez Martinez, Muharrem Akin, Vera Garcheva.

**Formal analysis:** L. Christian Napp, Carolina Sanchez Martinez, Muharrem Akin, Vera Garcheva, Christian Kühn, Johann Bauersachs, Andreas Schäfer.

**Writing – original draft:** L. Christian Napp, Christian Kühn, Andreas Schäfer.

**Writing – review & editing:** Carolina Sanchez Martinez, Muharrem Akin, Vera Garcheva, Johann Bauersachs.

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
