## [Decision Letter · Decision Letter 0]

3 Aug 2020

PONE-D-20-22896

Use of extracorporeal membrane oxygenation for eCPR in the emergency room in patients with refractory out-of-hospital cardiac arrest

PLOS ONE

Dear Dr. Schäfer,

Thank you for submitting your manuscript to PLOS ONE. After careful consideration, we feel that it has merit but does not fully meet PLOS ONE’s publication criteria as it currently stands. Therefore, we invite you to submit a revised version of the manuscript that addresses the points raised during the review process.

We look forward to receiving your revised manuscript.

Kind regards,

Gaetano Santulli

Academic Editor

PLOS ONE

Journal Requirements:

2. Thank you for including your ethics statement: 'The HAnnover Cooling REgistry (HACORE) is a prospective observational registry approved by the ethics committee at Hannover Medical School (#3567-2017) in accordance with the Declaration of Helsinki.'

a. Please amend your current ethics statement to confirm that your named institutional review board or ethics committee specifically approved this study.

"AS has received lecture and proctoring honoraria from Abiomed and ZOLL, LCN has received lecture honoraria from Maquet, Abbott, Zoll and Orion as well as lecture, consulting and proctoring honoraria from Abiomed. All other authors declare no conflict of interest."

Reviewers' comments:

Reviewer's Responses to Questions

**Comments to the Author**

1. Is the manuscript technically sound, and do the data support the conclusions?

Reviewer #1: Yes

Reviewer #2: Yes

2. Has the statistical analysis been performed appropriately and rigorously? 

Reviewer #1: Yes

Reviewer #2: Yes

3. Have the authors made all data underlying the findings in their manuscript fully available?

Reviewer #1: Yes

Reviewer #2: Yes

4. Is the manuscript presented in an intelligible fashion and written in standard English?

Reviewer #1: Yes

Reviewer #2: Yes

5. Review Comments to the Author

Reviewer #1: It was very refreshing to read your well-written manuscript that comes to a conclusion that one probably did not hope for. A "Hail Mary" treatment option such as eCPR is extremely difficult to implement in a timely fashion and adherence to selection criteria seems next to impossible. As you astutely mentioned, statistics do not apply to individuals, which explains why, in the heat of the moment, physicians may have difficulty adhering to such rigid criteria, not to mention that some of this information may not have been immediately available when the decision was made to consider eCPR. Your conclusion about needing to shorten low-flow time in the future paves the way for future eCPR studies and implementation. Finally, your tables are excellent, and it was especially interesting to read about the individual characteristics of the survivors in Table 3.

I have a few questions. Why do you suppose that 75% of the patients were male? Also, could you describe "mechanical compressions?" Do all of the ambulances in your region use Lucas or other devices? According to Table 3, the only survivor with a CPC score of 1 in your study did not have mechanical compressions until arriving in the ED. Also, could you please define the parameters of neuroprotective hypothermia?

The remainder of my comments are purely minor typographical errors. Lines 62-63: commas after numerous and younger. Line 63: spell out "minutes." Line 64: remove comma. Line 74: comma after 10%. Line 79: comma after here. Line 80: comma after selection. Remove comma after approach. Line 114: period after CPC. Line 176:...eCPR is obviously a very... Line 179: remove comma. Line 206 does not make sense. It should read more like: "Positive outcomes from eCPR are still limited, and only a few patients finally survive until discharge."

Reviewer #2: This is a nicely written paper examining clinical outcomes of e-CPR in 40 patients, showing 5 survivors with only 3 of those 5 having a good neurological outcome. I think the data are well presented and represent a large body of experience with this approach. With a growing use of ECMO this is an important topic. However the results were not so good and were consistent with previous publications. The authors point out why this is in that the ideal criteria were not applied to these patients. I think this paper could be greatly improved by adding a Table with a list of proposed criteria based on the experiences garnered from this data set. In other words who should we not put on bypass. This is a very important question in this data set is uniquely positioned to be able to suggest reasonable criteria. Perhaps the authors could do a speculative analysis of the data set that would suggest a set of criteria that might results in a survival rate of say 20%. In other words, keep the survivors, but identify a set of criteria that would reduce the non-survivor numbers to the point where the survival rate was 20% (or whatever % the authors feel would make eCPR a reasonable option from a chance of survival standpoint as well as cost. Also they should consider adding cost data.

6. PLOS authors have the option to publish the peer review history of their article (what does this mean?). If published, this will include your full peer review and any attached files.

Reviewer #1: **Yes: **Gayle Galletta, MD

Reviewer #2: **Yes: **T. Sloane Guy, MD

---

## [Author Response · Author response to Decision Letter 0]

11 Aug 2020

Response to Editor Manuscript PONE-D-20-22896

We would like to thank the Editor for his/her constructive criticism and valuable input. We hope that the changes made to the manuscript and the responses detailed below, satisfactorily address the concerns raised.

The manuscript has been formatted according to the recommendations given prior to any responses to the reviewers.

2. Thank you for including your ethics statement: 'The HAnnover Cooling REgistry (HACORE) is a prospective observational registry approved by the ethics committee at Hannover Medical School (#3567-2017) in accordance with the Declaration of Helsinki.'

a. Please amend your current ethics statement to confirm that your named institutional review board or ethics committee specifically approved this study.

The ethics statement has been amended as requested in the manuscript (lines 79-80) and the online submission form.

Written informed consent was obtained from legal guardians during the unconscious period and re-consented by survivors after gaining consciousness. Anonymized data were entered into the database. This statement is now included in the Methods section (lines 79-82) and the online submission from.

"AS has received lecture and proctoring honoraria from Abiomed and ZOLL, LCN has received lecture honoraria from Maquet, Abbott, Zoll and Orion as well as lecture, consulting and proctoring honoraria from Abiomed. All other authors declare no conflict of interest." Please confirm that this does not alter your adherence to all PLOS ONE policies on sharing data and materials, by including the following statement: "This does not alter our adherence to PLOS ONE policies on sharing data and materials.” 

The statement has been included in the “conflict of interest” section on page 18 as requested.

Regarding data availability there is a legal restriction as the smaller patient sample includes potentially identifying patient information and specific dates. Contact information for the ethics committee is: Ethikkommission@mh-hannover.de. This has been included in the revised cover letter.

Response to Reviewer #1 Manuscript PONE-D-20-22896

We would like to thank the reviewer for his/her constructive criticism and valuable input. We hope that the changes made to the manuscript and the responses detailed below, satisfactorily address the concerns raised.

It was very refreshing to read your well-written manuscript that comes to a conclusion that one probably did not hope for. A "Hail Mary" treatment option such as eCPR is extremely difficult to implement in a timely fashion and adherence to selection criteria seems next to impossible. As you astutely mentioned, statistics do not apply to individuals, which explains why, in the heat of the moment, physicians may have difficulty adhering to such rigid criteria, not to mention that some of this information may not have been immediately available when the decision was made to consider eCPR. Your conclusion about needing to shorten low-flow time in the future paves the way for future eCPR studies and implementation. Finally, your tables are excellent, and it was especially interesting to read about the individual characteristics of the survivors in Table 3.

We thank the reviewer for this positive recognition of our work.

I have a few questions. Why do you suppose that 75% of the patients were male? 

The majority of eCPR patients without obvious non-cardiac cause of arrest such as drowning had critical coronary stenosis suggestive for acute myocardial infarction. In a recent AMI-registry at our institution of almost 300 AMI patients, 77% were male. Therefore, considering AMI as the major cause for cardiac arrest in the eCPR population as well, a ratio about 75% male patients would not be too surprising. A statement regarding this has been added to the results section (lines 130-133).

Also, could you describe "mechanical compressions?" Do all of the ambulances in your region use Lucas or other devices? According to Table 3, the only survivor with a CPC score of 1 in your study did not have mechanical compressions until arriving in the ED. 

In order to prevent insufficient circulation during transport, we recommended the use of mechanical compression devices. In Germany, a rendezvous system consisting of a paramedic-staffed ambulance and an emergency physician-staffed response car or rescue helicopter are dispatched for cardiac arrest patients. In our city, all physician-staffed response cars and the rescue helicopters are equipped with the LUCAS device. The one patient mentioned by the reviewer was brought to our cardiac arrest centre by an anesthesiologist from our hospital, who was on call on a non-LUCAS-equipped response car outside the city boundaries. 

Also, could you please define the parameters of neuroprotective hypothermia?

As suggested by the reviewer, we have added a short description of mode of cooling, target temperature, duration of hypothermia, rewarming, and maintained normothermia to the methods section (lines 108-113).

The remainder of my comments are purely minor typographical errors. Lines 62-63: commas after numerous and younger. Line 63: spell out "minutes." Line 64: remove comma. Line 74: comma after 10%. Line 79: comma after here. Line 80: comma after selection. Remove comma after approach. Line 114: period after CPC. Line 176:...eCPR is obviously a very... Line 179: remove comma. 

We thank the reviewer for pointing out the typographical errors which have all been corrected in the revised version of the manuscript.

Line 206 does not make sense. It should read more like: "Positive outcomes from eCPR are still limited, and only a few patients finally survive until discharge."

We thank the reviewer for this helpful suggestion which has been adopted (lines 234-235). 

Response to Reviewer #2 Manuscript PONE-D-20-22896

We would like to thank the reviewer for his/her constructive criticism and valuable input. We hope that the changes made to the manuscript and the responses detailed below, satisfactorily address the concerns raised.

This is a nicely written paper examining clinical outcomes of e-CPR in 40 patients, showing 5 survivors with only 3 of those 5 having a good neurological outcome. I think the data are well presented and represent a large body of experience with this approach. With a growing use of ECMO this is an important topic. However the results were not so good and were consistent with previous publications. The authors point out why this is in that the ideal criteria were not applied to these patients. 

We thank the reviewer for this positive recognition of our work.

I think this paper could be greatly improved by adding a Table with a list of proposed criteria based on the experiences garnered from this data set. In other words who should we not put on bypass. This is a very important question in this data set is uniquely positioned to be able to suggest reasonable criteria. Perhaps the authors could do a speculative analysis of the data set that would suggest a set of criteria that might results in a survival rate of say 20%. In other words, keep the survivors, but identify a set of criteria that would reduce the non-survivor numbers to the point where the survival rate was 20% (or whatever % the authors feel would make eCPR a reasonable option from a chance of survival standpoint as well as cost. 

The reviewer’s point is very interesting. With all limitations that apply to such an approach, age >60 years, pulseless electrical activity and absence of bystander CPR could be used in our cohort to enrich the treated population towards a survival rate of 22%. We have added that hypothetical calculation as well as critical statement on influence of cost to the discussion section (lines 216-226).

Also they should consider adding cost data.

While cost data are certainly an interesting aspect, they differ enormously between countries and different health care systems. In some European countries the hospital has to pay for the use of ECMO, in some countries only excess costs are reimbursed, and in others the hospitals gain revenue by using extracorporeal devices. As appealing as this point is, we, therefore, restrained from adding dedicated cost data.

---

## [Decision Letter · Decision Letter 1]

14 Sep 2020

Use of extracorporeal membrane oxygenation for eCPR in the emergency room in patients with refractory out-of-hospital cardiac arrest

PONE-D-20-22896R1

Dear Dr. Schäfer,

We’re pleased to inform you that your manuscript has been judged scientifically suitable for publication and will be formally accepted for publication once it meets all outstanding technical requirements.

Kind regards,

Gaetano Santulli, MD

Academic Editor

PLOS ONE

Additional Editor Comments (optional):

Reviewers' comments:

Reviewer's Responses to Questions

**Comments to the Author**

1. If the authors have adequately addressed your comments raised in a previous round of review and you feel that this manuscript is now acceptable for publication, you may indicate that here to bypass the “Comments to the Author” section, enter your conflict of interest statement in the “Confidential to Editor” section, and submit your "Accept" recommendation.

Reviewer #1: All comments have been addressed

2. Is the manuscript technically sound, and do the data support the conclusions?

Reviewer #1: Yes

3. Has the statistical analysis been performed appropriately and rigorously? 

Reviewer #1: I Don't Know

4. Have the authors made all data underlying the findings in their manuscript fully available?

Reviewer #1: Yes

5. Is the manuscript presented in an intelligible fashion and written in standard English?

Reviewer #1: Yes

6. Review Comments to the Author

Reviewer #1: (No Response)

7. PLOS authors have the option to publish the peer review history of their article (what does this mean?). If published, this will include your full peer review and any attached files.

Reviewer #1: **Yes: **Gayle Galletta, MD

---

## [Editor Report · Acceptance letter]

17 Sep 2020

PONE-D-20-22896R1 

Use of extracorporeal membrane oxygenation for eCPR in the emergency room in patients with refractory out-of-hospital cardiac arrest 

Dear Dr. Schäfer:

I'm pleased to inform you that your manuscript has been deemed suitable for publication in PLOS ONE. Congratulations! Your manuscript is now with our production department. 

Kind regards, 

on behalf of

Prof. Gaetano Santulli 

Academic Editor

PLOS ONE